# Comparison of Oxidative and Physical Stabilities of Conjugated Linoleic Acid Emulsions Stabilized by Glycosylated Whey Protein Hydrolysates via Two Pathways

**DOI:** 10.3390/foods11131848

**Published:** 2022-06-23

**Authors:** Meng Li, Jinzhe Li, Yuxuan Huang, Munkh-Amgalan Gantumur, Akhunzada Bilawal, Abdul Qayum, Zhanmei Jiang

**Affiliations:** Key Laboratory of Dairy Science (Ministry of Education), College of Food Science, Northeast Agricultural University, Harbin 150030, China; lm18745783719lm@163.com (M.L.); a159846a@163.com (J.L.); b6589634b@163.com (Y.H.); a584138a@163.com (M.-A.G.); a4859884a@163.com (A.B.); abqyum7@gmail.com (A.Q.)

**Keywords:** whey protein isolate, conjugated linoleic acid, emulsion, glycosylated hydrolysates, oxidative stability, physical stability

## Abstract

The objective of the research was to analyze and compare the oxidative and physical stabilities of conjugated linoleic acid (CLA) emulsions stabilized by two glycosylated hydrolysates (GPP-A and GPP-B) that were formed via two different pathways. This study showed that GPP-A exhibited higher browning intensity and DPPH radical scavenging ability in comparison with GPP-B. Moreover, the CLA emulsion formed by GPP-A exhibited a lower creaming index, average particle size, primary and secondary oxidative products, in comparison with GPP-B-loaded emulsion. However, the GPP-A-loaded emulsion showed a higher absolute potential and fraction of interfacial adsorption than that of the CLA emulsion formed by GPP-B. Therefore, the CLA emulsion formed by GPP-A exhibited stronger stabilities in comparison with the GPP-B-loaded emulsion. These results suggested that GPP-A showed an emulsification-based delivery system for embedding CLA to avoid the loss of biological activities. Additionally, the development of CLA emulsions could exert its physiological functions and prevent its oxidation.

## 1. Introduction

Conjugated linoleic acid (CLA) is a mixture of positional and geometric isomers of linoleic acid. It is a polyunsaturated fatty acid with various physiological activities, including promoting lipid metabolism, improving immune function, reducing inflammation, inhibiting the growth of cancer cells, and so on. However, CLA is poorly soluble in water and easily decomposed by light, heat and oxygen, thus limiting its applications in the food field. Therefore, establishing a stable system for the application of CLA in the food industry is very necessary. Moreover, microcapsules and emulsions are common homeostasis techniques [1]. Currently, CLA, as a bioactive oil, has been widely used in emulsions as a steady-state technology.

Oil-in-water (O/W) emulsions are widely used in the food industry, such as in milk, salad cream, ice cream, and so on. Generally, synthetic and natural emulsifiers are used to stabilize the emulsions. A study has found that low molecular weight synthesis could be used as a synthetic emulsifier, which could improve the adsorption ability of the oil–water surface, thus enhancing the emulsifying activity [2]. However, synthetic emulsifiers may potentially harm human health, thereby limiting their application in food. Therefore, macromolecular proteins, with a high nutritional value and good emulsification, have been widely used to stabilize emulsions in recent years. However, many studies showed that the emulsions formed by untreated protein did not form a dense adsorption layer around the droplets, thus decreasing the stability of the emulsions during storage [3,4]. Additionally, polyunsaturated fatty acids in O/W emulsions are easily affected by oxygen, heat, light, enzymes, metals and metalloproteins, which lead to lipid oxidation of the emulsions, thus effecting the texture, flavor, appearance and shelf life of emulsions. Accordingly, it is very important to develop an emulsifier with strong antioxidant property to stabilize O/W emulsions. Importantly, O/W emulsions can be used to encapsulate probiotics or bioactive substances, thus exerting their biological activity [5,6,7].

Recently, some studies have proved that the functional properties of glycosylated proteins were significantly enhanced, compared with that of untreated proteins [8]. Meanwhile, the glycosylated products as emulsifiers were more easily adsorbed on the oil–water interface, thus enhancing the physical stability of O/W emulsions. A study suggested that citral-loaded emulsions formed by soy protein–soy polysaccharose showed a lower average particle size than soy protein-loaded emulsions [9]. Moreover, oat protein and dextran conjugate-stabilized emulsions exhibited better storage stability and more uniform oil droplets under different environmental conditions, in comparison with the O/W emulsions formed by oat protein [10]. Moreover, hydrolyzed proteins have been widely used because of their mild reaction conditions, simple operation, few by-products and high specificity. Additionally, protein hydrolysates have potential ability as emulsifiers, thus improving the oxidative stability of emulsions formed by hydrolysates [8]. For example, corn emulsions formed by soy protein peptides exhibited a lower average particle size and creaming index in comparison with the soy protein-loaded emulsions [11].

In addition, glycosylation and hydrolysis could change the structure of protein, thereby ameliorating the antioxidant and emulsifying characteristics. Therefore, the combination of the glycosylation and hydrolysis used for protein modification had recently attracted extensive attention. It was shown that corn emulsions stabilized by glycosylated conjugate, formed by pea protein hydrolysates and gum Arabic, exhibited stronger physical and oxidative stabilities, compared with pea protein hydrolysates-loaded emulsions [10]. Our previous study found that glycosylated whey-protein hydrolysate-loaded CLA emulsions showed higher physical and oxidative stability, compared with the emulsions formed by whey protein isolate [12]. Furthermore, glycosylated whey-protein hydrolysates could be prepared via two different pathways. During food processing, whey-protein hydrolysate was glycosylated and the obtained glycosylated whey-protein hydrolysate A was labelled as GPP-A, whereas the glycosylated whey-protein was hydrolyzed and the formed glycosylated whey-protein hydrolysate B was labelled as GPP-B. Moreover, in our previous study, it was also proved that there were significant differences in the characteristics between GPP-A and GPP-B [13]. However, the oxidative and physical stabilities of emulsions stabilized by glycosylated hydrolysates via two pathways have not been yet studied clearly.

The purpose of this study was to compare the stabilities of the CLA emulsions stabilized by glycosylated hydrolysate A (GPP-A) and glycosylated hydrolysate B (GPP-B). Initially, the browning intensity, free amino production and antioxidant activity of GPP-A and GPP-B were compared. Moreover, the CLA emulsions stabilized by glycosylated hydrolysates via the two pathways were measured to analyze their oxidative stabilities, involving primary and secondary oxidative products. Furthermore, the physical stability of the emulsions formed by GPP-A and GPP-B were analyzed, including the average particle size, ζ-potential, creaming index, microscopic images, contact angle and interfacial tension. Therefore, this work would provide the best pathway for glycosylate peptides, which could be applied to form a more stable emulsion to deliver CLA in the food field.

## 2. Materials and Methods

### 2.1. Experimental Materials

The whey protein isolate (WPI) (93% purity) was obtained from Mullins Whey Co., Ltd. (Mosinee, WI, USA). The other reagents were obtained from Sigma Co. Ltd. (St. Louis, MO, USA). Furthermore, the purity of the food-grade CLA was 80%.

### 2.2. Preparation of Two Glycosylated Hydrolysates

#### 2.2.1. Preparation of Glycosylated Hydrolysate A

The preparation of the two glycosylated hydrolysates was completed according to the authors of [12]. The WPI solution (3%, *w*/*v*) provided a pH adjustment of 7.0. Then, AS 1.398 neutral protease (E/S = 5%) was added to the mixture and hydrolyzed in the water bath at 45 °C. During the hydrolysis process, 1 mol/L NaOH was continuously used to maintain pH of the mixture at 7.0. After hydrolysis for 120 min, the mixture was placed in the water bath at 80 °C for 10 min to inactivate the enzyme. Subsequently, galactose (3%, *w*/*v*) was added to the hydrolysate to prepare the glycosylated mixture. The glycosylated mixture was provided a pH adjustment of 8.0. Then the mixture was bathed at 90 °C for 4 h to prepare the glycosylated hydrolysate A (GPP-A).

#### 2.2.2. Preparation of Glycosylated Hydrolysate B

The glycosylated mixture was prepared by adding galactose (3%, *w*/*v*) to the WPI solution (3%, *w*/*v*). The glycosylated products were prepared according to the method set out in Section 2.2.1. Subsequently, the glycosylated product had a pH adjustment of 7.0 by using 1 mol/L NaOH. The glycosylated hydrolysate B (GPP-B) was produced according to the method set out in Section 2.2.1.

### 2.3. Preparation of Emulsions Stabilized by Two Glycosylated Hydrolysates

The CLA was added to the GPP-A and GPP-B at a ratio of 1:9, respectively. Then, pre-emulsions were prepared by a high-speed disperser (10,000 rpm) for 4 min. Subsequently, the CLA emulsions were prepared by using a high-pressure homogenizer for six cycles at 80 MPa. Moreover, the CLA emulsion stabilized by the WPI was used as the blank control group.

### 2.4. Browning Intensity

The determination of the browning intensity was according to the described method in [14]. The GPP-A and GPP-B were diluted to 6 mg/mL, and the absorbance of the diluted solutions was determined at 420 nm.

### 2.5. Free Amino Production

The free amino production was measured according to the method of OPA [15]. The OPA reagent should be used and prepared on the spot. The diluent of the samples and OPA reagent were mixed and reacted immediately without light for 5 min. Then, the absorbance of the mixture was measured at 340 nm. According to the above method, L-leucine was used as the standard to draw the standard curve Y = 1.667 x + 0.1397, R2 = 0.9991 (the consistence as the *X*-axis; the absorbance as the *Y*-axis), thus calculating the free amino content.

### 2.6. Antioxidant Property

The diluent of the samples (15 mg/mL) and DPPH solution (0.1 mM) were mixed and reacted immediately without light for 30 min. Then, the absorbance of the mixture was measured at 517 nm. The DPPH radical scavenging rate was calculated, according to the authors of [16].

### 2.7. Oxidative Stabilities

The peroxide value (POV) and thiobarbituric acid-reactive substances (TBARS) were analyzed by measuring the primary and secondary oxidative products of the emulsions [13]. The absorbance of the samples was measured at 510 and 532 nm to analyze POV and TBARS, respectively. The oxidative stabilities of the CLA emulsions were measured when stored for 0, 3, 6, 9, 12 and 15 days.

### 2.8. Mean Particle Size and ζ-Potential

The average particle size and ζ-potential of the CLA emulsions were determined by dynamic light scatter meter, according to the authors of [17]. The mean particle size and ζ-potential were determined based on the principles of dynamic light scattering and microelectrophoresis.

### 2.9. Creaming Index

According to the authors of [18], 8 mL of emulsion was placed in the sample bottles at 25 °C. Meanwhile, the stratification of the CLA emulsions was observed when stored for 0, 3, 6, 9, 12 and 15 days. The CI was determined as follows:CI(%) = H_C_/H_S_ × 100%(1)
H_C_ was the height (cm) of the creaming volume of CLA emulsions;H_S_ was the total height (cm) of emulsion samples.

### 2.10. Fraction of Interfacial Adsorption

The fraction of interfacial adsorption (Fads) was determined according to the authors of [19]. The emulsion samples were centrifuged at 15,000 rpm for 60 min, then the subnatant was absorbed and filtered by 0.22 μm filter. Subsequently, the interfacial protein content was determined by the Coomassie bright blue method.

### 2.11. Confocal Laser Scanning Microscopy

Confocal laser scanning microscopy (CLSM) was applied to analyze the microscopic structure of the CLA emulsions, according to the authors of [20]. Nile red and Nile blue were dissolved in an isopropyl alcohol solution, and then stored without light for 30 min and filtered through a 0.45 μm filter. The proteins were dyed with Nile blue, and the CLA was dyed with Nile red for fluorescence staining. Then, the microstructure of the stained emulsion was observed.

### 2.12. Contact Angle and Interfacial Tension

The contact angle and interfacial tension were determined, according to the authors of [21] Preparation of amphiphilic Janus SiO_2_ particles and, by using an Optical Contact Angle Measuring Device. The CLA emulsion samples were absorbed with a syringe, then pressing the syringe and dropping one drop of emulsion (5 μL) on the slide. The droplet was photographed with a camera immediately after it began to fall and the interfacial tension curve was plotted over time.

### 2.13. Statistical Analysis

All of the experiments were performed in triplicate. The significant differences were analyzed using statistical software of SPSS 20.0 with Duncan’s test at *p* < 0.05.

## 3. Results

### 3.1. Comparison of Characterizations of Glycosylated Hydrolysates

The browning intensity, free amino production and antioxidant properties are shown in Figure 1A–C to assess the physicochemical properties of the GPP-A and GPP-B. It was shown that two factors can be used to evaluate the degree of glycosylation: browning intensity and free amino content [22]. Meanwhile, the antioxidant activity of glycosylated hydrolysates via two pathways was determined by the DPPH radical scavenging rate in this study.

The absorbance at 420 nm of browning intensity is an important indicator for judging the advanced stage of glycosylation. Figure 1A suggested that the absorbance of the WPI, GPP-A and GPP-B was 0.326, 0.718 and 0.423, respectively. It demonstrated that the browning intensity of the GPP through the two pathways was significantly higher than that of the WPI (*p* < 0.05). There were unsaturated brown substances (such as copolymers, melanonids, nitrogenous polymers) generating during the final stage of the glycosylated reaction [23]. It suggested that glycosylation formed by the porcine-plasma protein hydrolysates and the three kinds of monosaccharides had a deeper color, compared with the porcine-plasma protein hydrolysates [24]. Meanwhile, the absorbance of GPP-A was 70.73% higher than that of GPP-B, indicating that GPP-A exhibited stronger glycosylation development. This indicated that the glycosylated hydrolysates via pathway A could form more unsaturated brown substances, causing a stronger browning intensity. A similar result was provided showing that the UV-absorbance of hydrolyzed glycosylated products was higher than glycosylated hydrolysates [25].

From Figure 1B, the free amino production of the WPI was significantly lower than that of GPP-A and GPP-B (*p* < 0.05). It suggested that the amino N of the proteins was cross-linked with the carbonyl of galactose. The glycosylated conjugation formed by the hydrolysates of zein and chitosan showed a higher free amino content in comparison to zein [26]. Moreover, the hydrolysates of soy protein-maltodextrin consumed more free amino groups in comparison to soy protein [27]. Furthermore, the free amino content of GPP-B was significantly lower than that of GPP-A (*p* < 0.05), exhibiting that the pathway A could consume more free amino groups, compared with pathway B. Therefore, the reactivity of GPP-A was higher, which was consistent with the consequence of the browning intensity (Figure 1A). The authors of [25] exhibited that the molecular weight of GPP-A was lower than that of GPP-B, thus consuming more of the free amino groups of GPP-B.

Figure 1C indicates that the antioxidant properties of the WPI, GPP-A and GPP-B were 22.77, 93.37 and 84.18%, respectively. It indicated that the glycosylated hydrolysates generated by both of the two pathways could significantly increase the radical scavenging rate in comparison with the WPI (*p* < 0.05). Moreover, the GPP-A showed stronger antioxidant properties than GPP-B. A study showed that the glycosylation reaction could produce melanosomes with strong antioxidant activities, resulting in enhancing the DPPH radical scavenging rate [28]. Moreover, there were more peptides with strong antioxidant properties generating after hydrolysis [11]. It has been demonstrated that the glycosylated products formed by the hydrolysates of chicken breast meat and xylose show stronger antioxidant activities, when compared with chicken breast meat [29]. Furthermore, the GPP-A exhibited a higher DPPH radical scavenging rate than the GPP-B in Figure 1C. It indicated that pathway A had stronger free radical scavenging activity, thus showing a stronger hydrogen supply ability. Additionally, the DPPH radical scavenging activity of the hydrolyzed glycosylated product also increased with the increase of hydrolysis time, due to a reduction in its molecular weight [30]. Therefore, the GPP-A with a lower molecular weight exhibited higher DPPH radical scavenging activity.

### 3.2. Comparison of Oxidative Stability of Emulsions Stabilized by Glycosylated Hydrolysates

During the storage of the emulsions, oil oxidation might cause rancidity, deterioration and loss in quality of foods; therefore, improving the oxidative stabilities of emulsions is an urgent problem to be solved. In this study, the primary and secondary oxidative products of the CLA emulsions were analyzed by measuring the POV and TBARS at 25 °C, shown in Figure 2A,B, respectively.

The POV and TBARS of all of the CLA emulsions exhibited a significant trend during the period of storage. From Figure 2A, the POV of GPP-A- and GPP-B-stabilized CLA emulsions decreased by 125.35 and 33.96%, respectively, after being stored for 15 days. In Figure 2B, initially, all of the CLA emulsions showed similar quantities of TBARS. Moreover, following storage for 15 days, the TBARS of the emulsions formed by the WPI, GPP-A and GPP-B were 1.36 mmol/kg, 0.57 mmol/kg and 0.77 mmol/kg. It suggested that the CLA emulsions stabilized by the two pathways had stronger oxidative stability, compared with the WPI-loaded emulsion. Because the DPPH radical scavenging rate of the GPP-A and GPP-B was higher than the WPI (Figure 1C), this led to stronger oxidative stability as emulsifiers. A previous study showed that the POV of astaxanthin emulsions formed by whey protein-flaxseed gum was lower than whey protein-loaded emulsion [31]. Moreover, the TBARS of fish oil emulsions formed by hydrolyzed soy protein-maltodextrin were lower than those of soy protein-loaded emulsion [27]. Moreover, the primary and secondary oxidative products of the CLA emulsions formed by GPP-A were significantly lower than the GPP-B-loaded emulsions after 15 days. It is suggested that GPP-A with stronger antioxidant properties absorbed at the O/W interface of emulsion droplets could improve the antioxidant activities, to reduce lipid oxidation and protect the CLA. Meanwhile, the oil–water interface layer formed by the GPP-A-stabilized emulsion could produce a dense physical barrier to protect the CLA. Therefore, the GPP-A-stabilized CLA emulsion had stronger oxidative stability in comparison with the GPP-B-loaded emulsion, thus inhibiting the lipid oxidation of the CLA.

### 3.3. Comparison of Physical Stability of Emulsions Stabilized by Glycosylated Hydrolysates

The mean particle size, ζ-potential and creaming index (CI) are exhibited in Figure 3A–C, respectively, which were used to analyze the physical stability of the CLA emulsions stabilized by GPP-A and GPP-B.

The mean particle size is the important index to analyze emulsion stabilities [32,33]. As shown in Figure 3A, the mean particle size of the emulsions formed by GPP-A and GPP-B was observably and significantly lower than that of the WPI-stabilized emulsion (*p* < 0.05). It indicated that the combination of glycosylation and hydrolysis to modify the WPI as emulsifiers could significantly reduce the mean particle size of the emulsions, thus improving the physical stability. From Figure 3B, the absolute potential of the CLA emulsions loaded by GPP-A and GPP-B increased by 133.13 and 74.23%, in comparison with that of the WPI-loaded emulsion. Furthermore, the GPP-A-stabilized emulsion had a higher absolute ζ-potential than the CLA emulsion loaded by GPP-B. It can be inferred, from Figure 3A,B, that the glycosylated WPI tended to adsorb on the oil–water interface, thus increasing the repulsive term and restraining the aggregation of emulsion droplets. Additionally, the hydrophobic groups of hydrolyzed protein were exposed, which could improve the surface hydrophobicity and the molecular flexibility of protein [12]. These were conducive to adsorption on the surface of the oil droplets, thus effectively reducing the interfacial tension, and forming a smaller particle size and more stable emulsions [30]. As shown in Figure 3C, the CI of the CLA emulsions stabilized by the WPI, GPP-A and GPP-B increased significantly with storage time (0–15 days). The CI of the emulsions loaded by the WPI and GPP-B was 82.6 and 8.37% when stored for 15 days, while there was no significantly cream layer forming in the GPP-A-loaded emulsion, from the inserted image of Figure 3D. These results indicated that the combination of glycosylation and hydrolysis could significantly reduce the CI of CLA emulsions. Furthermore, GPP-A as the emulsifier could more easily be adsorbed to the oil–water interface, thus inhibiting the aggregation of the oil droplets. It suggested that the emulsions formed by hydrolysates of the whey protein-linear dextrin showed a higher absolute ζ-potential and smaller average particle size than that of the whey protein-loaded emulsion [34].

### 3.4. Comparison of Fraction of Interfacial Adsorption of Emulsions Stabilized by Glycosylated Hydrolysates

The fraction of interfacial adsorption (Fads) is displayed in Figure 4. Generally, Fads is the key factor for the stability of emulsions and for preventing oil droplets from coalescing. From Figure 4, the Fads of the CLA emulsions formed by the WPI, GPP-A and GPP-B was 48.79, 89.24 and 68.17%, respectively. It indicated that the Fads of emulsions formed by glycosylated hydrolysates as emulsifiers through the two pathways were more than those of the WPI-loaded emulsion (*p* < 0.05). A similar result showed that the Fads of emulsions formed by a walnut protein isolate-glucose conjugate was significantly higher than a walnut protein isolate-loaded emulsion [35]. Additionally, the emulsions stabilized by the hydrolysates of egg-yolk protein had higher Fads in comparison with the egg-yolk protein-loaded emulsion [36]. Meanwhile, the CLA emulsion formed by GPP-A had more Fads than pathway B as the emulsifier (*p* < 0.05). The GPP-A molecule was significantly expanded with a higher reaction degree, and more hydrophobic groups were exposed. A study suggested that the glycosylated products could wrap on the surface of the oil droplets, resulting in the improvement of Fads [37]. Moreover, after hydrolysis, the protein structure expanded and surface hydrophobicity and molecular flexibility were enhanced. Then, the hydrophobic peptides were more inclined to move to the non-water interface and attach to the surface of oil droplets, thus enhancing Fads [38].

### 3.5. Comparison of Microstructure of CLA Emulsions Stabilized by Glycosylated Hydrolysates

Figure 5 shows the microscopic structure of the CLA emulsions stabilized by the WPI, GPP-A and GPP-B, directly observed by confocal laser scanning microscopy (CLSM). All of the micrographs of the CLA emulsions showed that the oil droplets and protein were dyed in red and green, respectively, by Nile Red and Nile Blue. The microscopic images of the emulsion droplets exhibited the structure of red in green, which indicated that all of the emulsion samples were of an oil in water (O/W) structure. As shown in Figure 5, the emulsion droplets of the WPI were not homogeneous, and formed a large number of droplet aggregates. Additionally, parts of the emulsion droplets of the WPI appeared as yellow aggregates. There were not enough emulsifiers of the WPI-stabilized emulsion to cover the oil droplets, so the oil droplets (red) were covered with protein (green), resulting in a yellow color of the overlay image (green + red = yellow) [39]. By comparison, the micrographs of the CLA emulsions stabilized by GPP-A and GPP-B exhibited not only uniform droplets, but also showed much smaller droplets than the WPI-loaded emulsion. This might be that the glycosylated hydrolysates, via the two pathways, were more easily absorbed on the surface of the droplets, in comparison with the WPI-stabilized emulsion, thus improving steric hindrance and preventing droplet aggregation [40]. Furthermore, the emulsions stabilized by hydrolysates of soy conglycinin-dextran had more uniform and smaller droplets, compared with the soy conglycinin-dextran-loaded emulsion [41]. Furthermore, it was previously reported that the emulsions stabilized by glycosylation formed by shrimp hydrolysate and xylose showed a more uniform microstructure, compared with shrimp hydrolysate-loaded emulsion [1].

### 3.6. Comparison of Contact Angle of CLA Emulsions Stabilized by Glycosylated Hydrolysates

The properties of the gas–water interface of the emulsions can be analyzed by its static contact angle and interfacial tension, which is used to judge the physical stability of emulsion [37]. Figure 6A,B depicts the contact angle and interfacial tension of the WPI-, GPP-A-, GPP-B-loaded emulsions. The contact angle of all of the emulsion samples was less than 60 °C, indicating that all of the emulsions had great hydrophilic properties. Specifically, the contact angle of the WPI, GPP-A and GPP-B was 32.02, 8.49 and 11.65°, respectively. It suggested that the glycosylated hydrolysates via the two pathways could decrease the contact angle of the CLA emulsions, compared with the WPI-stabilized emulsion. Moreover, the interfacial tension of the emulsions stabilized by glycosylated hydrolysates was significantly decreased, compared with the WPI-loaded emulsion in the rage of 0–200 s (*p* < 0.05). This was because the glycosylated hydrolysates might allow protein molecules to unfold, thus making it easier to adsorb on the oil–water interface of the emulsions [42]. Furthermore, the contact angle of the emulsion stabilized by GPP-A decreased by 37.22% in comparison with GPP-B-loaded emulsion. The GPP-A-stabilized emulsion showed a lower interfacial tension than the emulsion formed by GPP-B (*p* < 0.05). It was suggested that GPP-A had a smaller molecular weight and higher reaction degree, exposing more hydrophobic groups. Additionally, as the protein was adsorbed, the protein molecules were already distributed at the oil–water interface, resulting in a significant decrease in interfacial tension [43].

## 4. Conclusions

The oxidative and physical stabilities of the CLA emulsions formed by glycosylated hydrolysate A (GPP-A) and glycosylated hydrolysate B (GPP-B) were initially compared in the present study. The GPP-A exhibited higher browning intensity and DPPH-radical scavenging ability in comparison with the GPP-B. The CLA emulsion formed by the GPP-A exhibited a lower creaming index, particle size and primary, secondary oxidative products. In addition, it had a higher absolute potential and fraction of interfacial adsorption than the GPP-B CLA emulsion. The GPP-A showed an emulsification-based delivery system for embedding CLA and it avoided the loss of biological activities.

## Figures and Tables

**Figure 1 foods-11-01848-f001:**
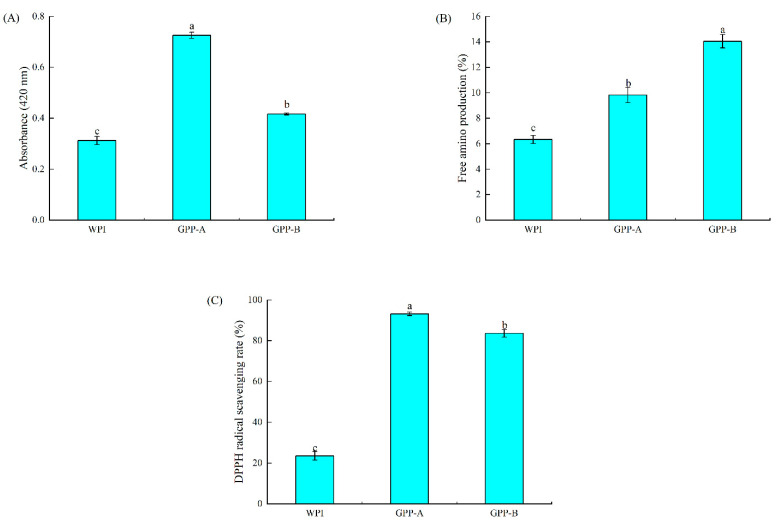
Changes in browning intensity (**A**); free amino production (**B**); and DPPH radical scavenge ability (**C**) of GPP-A and GPP-B. Values with different letters are significantly different (*p* < 0.05).

**Figure 2 foods-11-01848-f002:**
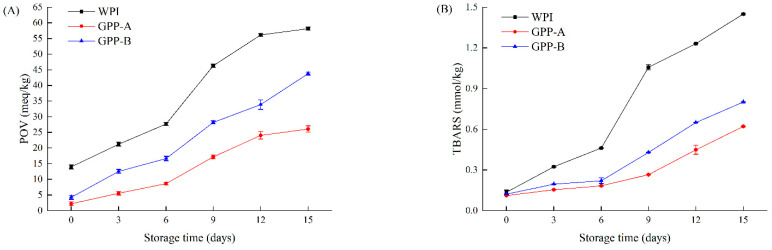
Changes in POV (**A**) and TBARS (**B**) of CLA emulsions stabilized by GPP-A and GPP-B during storage for 0–15 days at 25 °C.

**Figure 3 foods-11-01848-f003:**
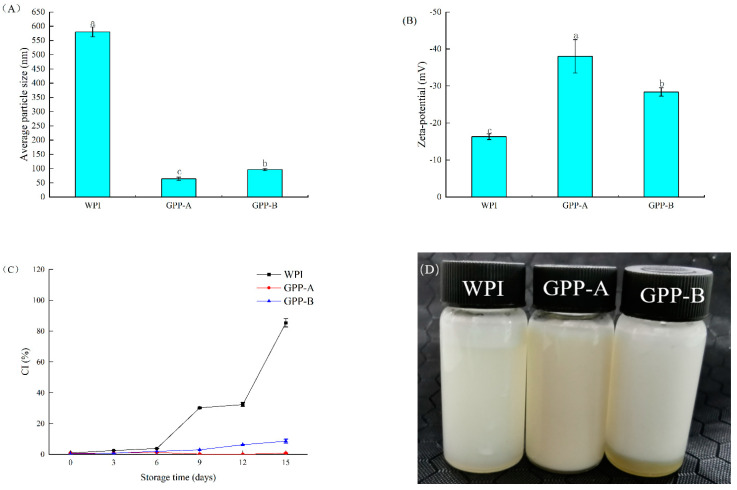
Changes in average particle size (**A**); zeta−potential (**B**); creaming index (**C**) and macroscopic pictures when stored for 15 days at 25 °C (**D**) of CLA emulsions stabilized by GPP-A and GPP-B. Values with different letters are significantly different (*p* < 0.05).

**Figure 4 foods-11-01848-f004:**
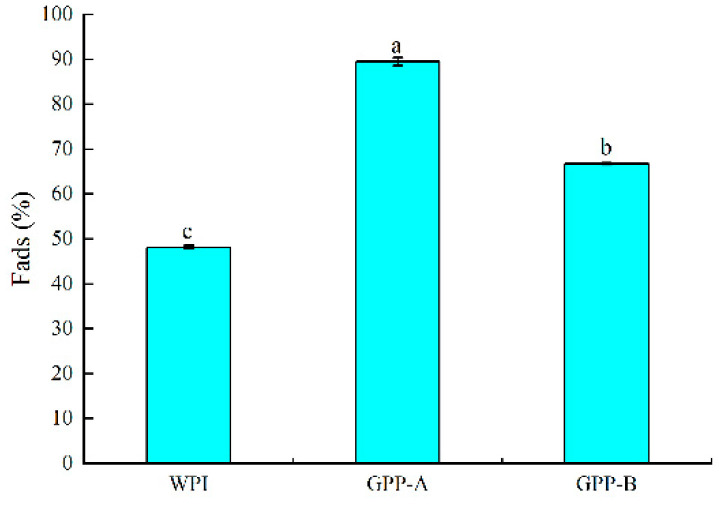
Changes in fraction of interfacial adsorption (Fads) of CLA emulsions stabilized by GPP-A and GPP-B. Values with different letters are significantly different (*p* < 0.05).

**Figure 5 foods-11-01848-f005:**
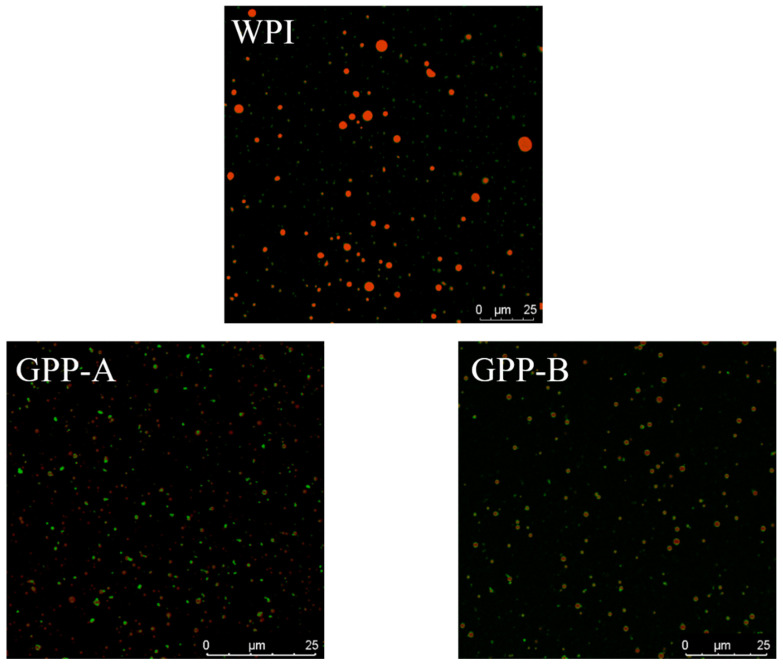
Changes in micrographs by using confocal laser scanning microscopy (CLSM) of CLA emulsions stabilized by GPP-A and GPP-B. All of the images were obtained with the Nile blue fluorescent dye. Magnification 40× and bar represent 50 µm in length.

**Figure 6 foods-11-01848-f006:**
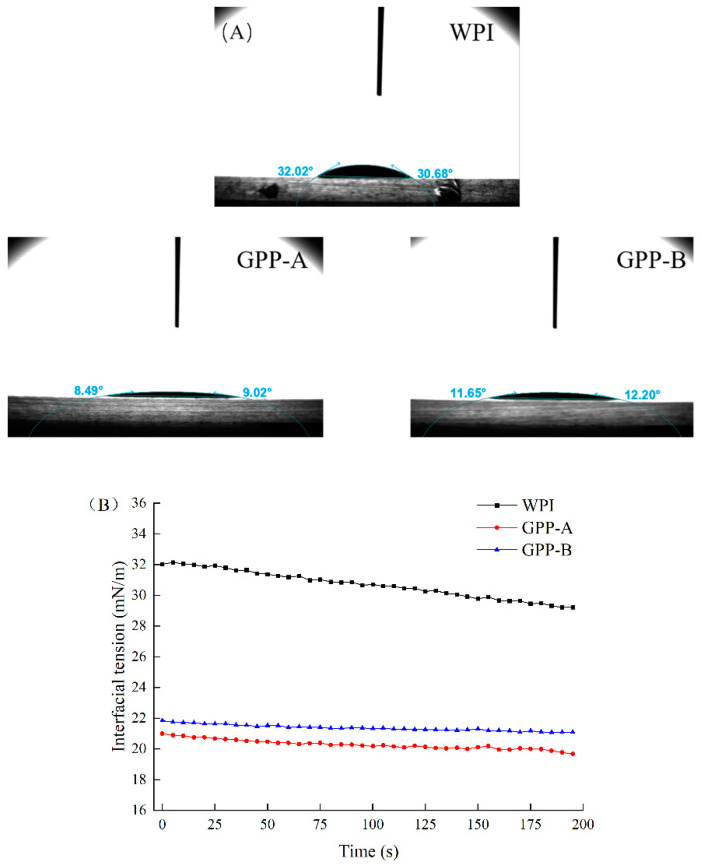
Changes in contact angle (**A**) and interfacial tension (**B**) of CLA emulsions stabilized by GPP-A and GPP-B.

## Data Availability

Data are contained within the article.

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
