# Peer review of "Comparison of Oxidative and Physical Stabilities of Conjugated Linoleic Acid Emulsions Stabilized by Glycosylated Whey Protein Hydrolysates via Two Pathways"

_foods, 2022, doi:10.3390/foods11131848_

Round 1
Reviewer 1 Report
This research was to analyze and compare oxidative and physical stabilities 1of conjugated linoleic acid (CLA) emulsions stabilized by two glycosylated hydrolysates (GPP-A 14 and GPP-B) that were formed via two different pathways
The conclusion of the manuscript must be improved.
The conclusion should have the answer for this question. Which does glycosylated hydrolysable has the best characteristics for a stability emulsion in formed concise.
Figure 3. ¿What time is the photography? and the photograph should be bigger.
Figure 4 should be centrated
Figure 5. The photography of GPP A and GPP B should be separated
The conclusion can be write “The GPP-A exhibited higher browning intensity and DPPH radical scavenge ability in comparison with GPP-B. The CLA emulsion formed by GPP-A exhibited lower creaming index, particle size and primary, secondary oxidative. Also it had a higher absolute potential, fraction of interfacial adsorption exhibited and stronger CLA. The GPP-A showed an emulsification-based delivery system for embedding CLA and it avoid the loss of biological activities”
Reviewer 2 Report
In this study Authors compared stabilities of CLA emulsions stabilized by glycosylated hydrolysate A (GPP-A) and glycosylated hydrolysate B (GPP-B). This study is a continuation of a previous research on glycosylated whey protein hydrolysate-loaded CLA emulsions. Glycosylated whey protein hydrolysates could be prepared via two different pathways. During food processing, whey protein hydrolysate was glycosylated and obtained glycosylated whey protein hydrolysate A was labelled as GPP-A, whereas glycosylated whey protein was hydrolyzed and the formed glycosylated whey protein hydrolysate B was labelled as GPP-B.
The paper is well written. The results are interesting and they are valuable extension of the previous work.
3.5. Comparison of microstructure of CLA emulsions stabilized by glycosylated hydrolysates
Why you did not use an image analysis to show the differences between GPP-A and GPP-B?
94 change this sentence
Quality of the figures is very poor...use original data and not the figures scans.
Round 2
Reviewer 1 Report
The authors did the corrections